# Resource Consumption and Remuneration Aspects in Navigated Screw Fixation Procedures with or without Additional Sacroplasty for Fragility Fractures of the Sacrum—A Prospective Clinical Study

**DOI:** 10.3390/jcm11206136

**Published:** 2022-10-18

**Authors:** Horst Balling, Boris Michael Holzapfel, Wolfgang Böcker, Joerg Arnholdt

**Affiliations:** 1Department for Spine Surgery and Traumatology, Orthopaedische Fachklinik Schwarzach, Dekan-Graf-Str. 2-6, 94374 Schwarzach, Germany; 2Center for Spine Surgery, Neckar-Odenwald-Kliniken gGmbH Buchen, Dr.-Konrad-Adenauer-Str. 37, 74722 Buchen, Germany; 3Department of Orthopaedics and Trauma Surgery, Musculoskeletal University Center Munich (MUM), University Hospital, LMU Munich, Marchioninistr. 15, 81377 Munich, Germany

**Keywords:** 3D-image guided, economy, navigated, osteosynthesis, procedural costs, sacral fragility fracture, reimbursement, resource consumption, sacroplasty, screw fixation

## Abstract

Surgical treatment for sacral fragility fractures using navigation-assisted screw fixation (NSF) is a modern, technically demanding procedure. Additional sacroplasty (ASP) has been shown to provide only insignificant clinical benefits for patients. This investigation highlights procedural economic aspects and evaluates results with regard to resource scarcity in order to be able to decide, whether ASP has a justification in NSF procedures beyond clinical aspects. From February 2011 to May 2017, all individuals with sacral fragility fractures surgically treated using 3D-fluoroscopy for NSF (*n* = 26) or NSF + ASP (*n* = 26) were enrolled. Outcome parameters were operative time, 3D-/2D-radiation dose, 2D-fluoroscopy time, material costs and reimbursement. In the two groups, a total of 52 individuals with 124 fragility fracture sites in sacral vertebrae I and II were surgically treated with similar numbers of screws inserted (*p* ≈ 0.679) requiring similar 3D- (*p* ≈ 0.546) and 2D-fluoroscopy radiation doses (*p* ≈ 0.236). In procedures with ASP, average 2D-fluoroscopy time (46.6 s vs. 32.7 s, *p* ≈ 0.004), and mean surgical duration (119 min vs. 96 min, *p* ≈ 0.011) were significantly longer. Mean implant costs (EUR 668.68 vs. EUR 204.34, *p* < 0.001), and reimbursement (EUR 8416.01 vs. EUR 6584.49, *p* ≈ 0.006) were significantly higher. Although comparison of costs and reimbursements indicated a positive financial balance, profitability was not confirmed, because financial expense for extended operative time prevented an economic advantage of procedures with ASP in this investigation. A formula was developed based on presented study data to allow similar economical decisions in other health care systems or institutions with differing resource costs.

## 1. Introduction

In clinical practice, sacral fragility fractures are increasingly recognized as a source of immobilization, mostly in the elderly affected by osteoporosis [1], but they are also found during pregnancy [2,3], or after repetitive stress and sporting activity in young individuals [4,5]. Immobilizing lower back pain coinciding with radiographic evidence of degenerative lumbar disease in elderly individuals often initially prevents causal attribution of complaints to underlying sacral fragility fractures, especially if no history of trauma was declared. Moreover, because symptoms are usually non-specific, clinicians are easily distracted from diagnosing these fractures. Anterior pelvic ring fractures, however, can be valuable radiographic hints for co-existing sacral fragility fractures [6], which are usually not visible on plain radiographs unless inadequate remodeling processes show widened fracture gaps or callus formation over time. Anyway, as soon as pelvic magnetic resonance imaging is performed, fracture sites are detectable with high sensitivity, especially in T1 or STIR (short tau inversion recovery) sequences [7,8]. Computed tomography (CT) is mandatory to classify fracture types and determine whether fracture lines reach neural foramina or the spinal canal, which might have implications on the choice of treatment.

Therapeutic options range from conservative management including pain and anti-osteoporotic medication to a portfolio of surgical treatment alternatives ranging from sacroplasty to osteosynthesis or lumbosacral hinge fixation with or without application of reinforcing bone cement [9,10]. Iliosacral screw fixation is most commonly performed to address acute traumatic instabilities of the posterior pelvic ring, especially in young individuals [11]. In elderly frail subjects with sacral fragility fractures, however, sacral cement augmentation is recommended to manage painful immobilization [12,13,14]. Whether affected individuals are sufficiently treated by sacral cement augmentation, by stabilizing screw osteosynthesis, or by a combination of these, is still discussed controversially. Clear recommendations for or against certain surgical therapies based on procedure-specific advantages and disadvantages are currently being developed [1]. Especially in geriatric trauma care, it is commonly accepted that short surgical duration, less invasive techniques, and limited radiation dose are important factors to keep perioperative trauma on a low level [15,16]. Moreover, these procedural parameters do not only have a medical, but also an economic impact, and may help to identify the more advantageous procedure in competing surgical strategies. A comparative analysis on clinical improvement and cost-effectiveness of different types of sacroplasty showed similar clinical effects but lower profits for the more elaborate treatment option [17]. On the other hand, sacroiliac fusion procedures were found more cost-effective to manage sacroiliac joint dysfunction than conservative treatment, as was reported in a recent systematic review and meta-analysis [18].

This prospective trial investigates procedure-specific differences in resource consumption focusing on operative time, radiation dose, costs and reimbursement based on a collective from a previous prospective single-center cohort study on clinical results after navigation-assisted screw fixation (NSF) with versus without additional sacroplasty (ASP) for the treatment of sacral fragility fractures [1]. The current trial examines the hypothesis that NSF is inferior to the combination of NSF + ASP regarding procedural profitability. This hypothesis was rejected, if the alternative hypothesis (“NSF is not economically inferior to NSF + ASP”) was confirmed (non-inferiority trial).

## 2. Materials and Methods

From October 2011 to May 2017, all patients with non-displaced or minimally displaced fragility fractures of the sacrum (type II fractures, according to Rommens et al. [19,20]) were included in the study. After application of preset inclusion and exclusion criteria (Table 1) remaining individuals with central or transforaminal fracture patterns according to Denis et al. [21] had been allocated to group I (NSF), exclusively, to avoid sacroplasty-associated complications by cement leakage into neural foramina or the spinal canal through fracture gaps.

All other individuals had been assigned to study groups using block randomization to finally obtain equal numbers of participants for both treatment methods. In group II (NSF + ASP), cement augmentation had only been performed for symptomatic fracture sites.

Data on surgical time from skin incision to wound closure were recorded during surgeries and retained in the clinic’s digital documentation system. Radiation dose data were provided by intraoperative fluoroscopy dose protocols. Procedural costliness was investigated based on regular implant and device prices (taxes included) regarding numbers and dimensions of screws, washers, guidewires, sacroplasty trocars, bone fillers and amount of bone cement. These prices might differ from actual purchase conditions. Prices for materials also needed in NSF procedures in identical amounts, e.g., sterile drapes, navigation reference pin for single-use, navigation markers (*n* = 8), etc., were not included in calculations.

For this prospective study, the CONSORT statement was followed. This monocentric study was performed in the “Orthopaedische Fachklinik Schwarzach”, Dekan-Graf-Str. 2-6, 94374 Schwarzach, Germany. Formal consent had been obtained from all participating individuals. The study protocol was approved by the initiating clinic’s medical ethics committee, in accordance with the WMA (World Medical Association) Declaration of Helsinki from 1964 and its later amendments. The trial was a posteriori registered in the German Clinical Trials Register (DRKS, study-ID: DRKS00026917).

### 2.1. Surgical Procedure

The combination of a 3D-fluoroscope and navigation unit (O-arm^®^, Surgical Imaging System, StealthStation S7 Surgical Navigation System, Medtronic Sofamor Danek, Memphis, TN, USA) had been used in all procedures for (temporary) image-guided placement of guidewires for subsequent iliosacral/trans-sacral cannulated screw insertion and ASP. Participants were under general anesthesia, and positioned prone on a radiolucent operating table (Maquet Getinge Group, Rastatt, Germany), before the O-arm gantry was installed for fluoroscopic 2D-imaging of the pelvis in anteroposterior, lateral, inlet, and outlet views. O-arm positions were stored for exact reproduction of fluoroscopic imaging. Exposed skin was disinfected, patient and O-arm gantry were covered with sterile drapes prior to skin incision. After percutaneous fixation of the navigation reference pin to the posterior iliac crest, an initial pelvic 3D-fluoroscopy scan was performed using the previously stored anteroposterior gantry position for navigational data acquisition during anesthesiological respiration arrest to avoid trunk movements and achieve highly accurate data sets for subsequent navigation. With obtained data, the navigation unit displayed a virtual pelvis on the StealthStation monitor for real-time image-guidance. With a navigated drill-guide up to four 2.8 mm × 450 mm, guidewires were preferentially placed trans-sacral under constant image-guidance control in sacral vertebrae I and II through a lateral transgluteal mini-open approach to the iliac wing. In procedures with ASP, additional guidewires were temporarily placed via short axis to the sacrum through separate stab incisions to direct hollow needles (Osteo-Introducer, Medtronic Sofamor Danek, Memphis, TN, USA, or 11G-Trocar, Vexim Rebalancing Spine, Balma, France, respectively) close to clinically symptomatic sacral fracture sites. A second collimated 3D-fluoroscopy scan was conducted to confirm correct guidewire and hollow needle positions. Required screw lengths were virtually determined on the navigation unit display. Cortical drilling (5.0 mm diameter) of iliac and sacral cortices along lateral guidewires was followed by insertion of cannulated screws (7.3 mm cannulated screws, DePuy Synthes Cannulated Screw System, Zuchwil, Switzerland, or 7.5 mm cannulated TIS^TM^-screws, Königsee Implantate GmbH, Allendorf, Germany, respectively) with or without washers using a cannulated screwdriver. For ASP, bone cement was administered via hollow needles under 2D-fluoroscopic control. After the removal of lateral guidewires and hollow needles, surgical results were controlled by a third collimated 3D-fluoroscopy scan. Detailed descriptions of the surgical technique have been published elsewhere [1,22].

### 2.2. Study Endpoints 

Endpoints were surgical time for performing procedures, 3D-, and 2D-radiation dose including 2D-fluoroscopy time. A procedural costliness-reimbursement analysis was conducted comparing mean expenditure for surgical resource consumption and average inpatient reimbursement in study groups based on the German implementation of the DRG (diagnoses-related groups) system.

### 2.3. Statistical Analysis

Values were expressed as mean and range of continuous variables, or absolute numbers of categorical variables. The Student’s *t*-test was used for comparing continuous variables following a normal distribution, and the *chi*^2^-test for categorical variables. Only if sample sizes were small (n < 10), or contingency tables showed a very unequal distribution, the Fisher’s exact test was employed. Comparing DRG-related basic reimbursements for sacral screw fixation procedures with/without cement augmentation revealed an extra-remuneration of EUR 2080 in procedures with ASP. Thus, the study was planned to detect a minimal economically important difference (MEID) of EUR 1800 in reimbursements between groups with a power of 95% assuming a standard deviation of EUR 1800. Thus, sample sizes were calculated as 26 for each group using OpenEpi software (Version 3.01, www.openepi.com, lastly accessed on 17 May 2015). Confidence intervals (CIs) were set at 95%, significance levels at 5%. SPSS 15.0.1 for Windows (SPSS Inc., Chicago, IL, USA) was used for analyses.

Procedural cost-effectiveness of NSF + ASP was confirmed, if the 95% CI of ASP-related extra-costs (comprising additional implant/device prices and expenses for extended operative duration based on operating room (OR) costs of EUR 65 per OR-minute) was below the upper margin of the 95% CI of ASP-related remuneration (=difference between reimbursements for NSF + ASP and NSF). Otherwise, economical non-inferiority of NSF was assumed.

## 3. Results

From October 2011 to May 2017, 52 individuals with 124 non-displaced sacral fractures had undergone surgery in form of NSF or NSF + ASP (Figure 1) after allocation to study groups. 

In advance, seven patients with similar radiological fracture morphologies had been excluded due to successful conservative treatment (*n* = 1), for receiving extended surgery because of additional pathologies of the lumbar spine (*n* = 4), or indicated revision surgery after previous bilateral iliosacral screw fixation (*n* = 1), or unilateral sacroplasty (*n* = 1). The flow diagram is depicted in Figure 2. 

Baseline conditions are given in Table 2 showing significant differences only for fracture type distribution.

Fractures in group I (NSF) were transalar (*n* = 17), transforaminal (*n* = 8), or central (*n* = 1). All individuals in group II (NSF + ASP) had transalar fracture patterns (*n* = 26). [A comprehensive data overview is given in Appendix A].

Procedural costs for inserted screws and sacroplasties performed are shown in Table 3.

Adverse events associated with ASP (cement extrusions from sacral vertebrae into neighboring tissues) were asymptomatic, and those associated with NSF (e.g., guidewire misplacement, subfascial hematoma formation) were unspecific. No device-related adverse events or persisting neurological sequelae were seen [1].


Study Endpoint Results (Table 4, Figure 3)


ASP prolonged surgical durations significantly by on average 23 min compared with sole NSF procedures (*p* ≈ 0.011).

In both groups, 3D-radiation dose was comparable (*p* ≈ 0.546). In procedures with additional ASP, 2D-radiation dose was higher without reaching significance (*p* ≈ 0.236). Comparison of 2D-fluoroscopy time, however, showed significantly higher values in NSF + ASP compared with NSF (*p* ≈ 0.004).

Although expenditure for iliosacral/trans-sacral screw osteosynthesis was similar in both groups (*p* ≈ 0.174), significantly higher average implant costs (*p* < 0.001) and reimbursements (*p* ≈ 0.006) were found in NSF + ASP. Average reimbursement differences *between* intervention groups amounted to EUR 1831.52 (95%—CI [1312.87, 2350.16]) and exceeded mean implant/device costs of ASP (EUR 428.91) by a difference of EUR 1402.61. Additive *procedural* costs for extended operative time-demand were calculated being 1495 EUR on average based on OR-costs of EUR 65 per OR-minute (Figure 4). Thus, the cost-effectiveness of NSF + ASP could not be confirmed under given conditions.

## 4. Discussion

This is the first investigation in the literature giving a detailed analysis of two competing surgical treatment options for non-displaced or minimally displaced fragility fractures of the sacrum in terms of operative time, procedural radiation dose, costs and reimbursement.

A previous report on *clinical* outcome parameters and complications confirmed therapeutic non-inferiority of NSF compared with NSF + ASP according to pain level decline, improvement of fracture-related disability, and duration of postsurgical recovery until discharge from inpatient treatment [1]. In none of these categories, differences between groups exceeded established minimal clinically important differences.

In contrast, distinct differences were noticed in the current evaluation of procedural resource consumption of navigated screw fixation procedures with versus without ASP, showing significant impact of ASP on operative time (*p* ≈ 0.011) with an average plus of 23 min surgical duration, corresponding to a mean procedural prolongation of 24%.

Since 3D-fluoroscopy scans were performed after identical surgical steps in both groups [(I) navigational data acquisition, (II) guidewire position control, (III) screw/cement position control], amount of 3D-radiation dose was not dependent on treatment mode. However, ASP was expected to result in considerably higher 2D-radiation doses compared with procedures without ASP, but statistical significance was not reached. This might be due to our habit of narrowing the fluoroscopy field to the sacral levels operated on (collimation), which limited ASP-related amounts of 2D-radiation dose considerably, resulting in only insignificantly higher 2D-radiation dose values in group II (NSF + ASP). Investigating fluoroscopy time, however, is purged from collimation effects, because here only the time of activating the X-ray tube is measured. This explains why 2D-fluoroscopy time was significantly longer in procedures with ASP, whereas 2D-radiation doses were only insignificantly higher.

The German implementation of the DRG system allows for higher reimbursement, if closed reduction and iliosacral/trans-sacral screw osteosynthesis is performed in combination with ASP. The difference in average reimbursement (EUR 1831.52) distinctly exceeded mean material costs of ASP (EUR 428.91) signaling a positive net remuneration of NSF + ASP procedures. Nevertheless, an average surgical time prolongation of 24% had to be considered compared with sole iliosacral/trans-sacral screw fixation procedures. Theoretically, ASP remains economically profitable, if the positive net remuneration (EUR 1402.61) can level off 23 min additional surgical time, meaning that a single OR-minute needed to be available at a price below EUR 60.98. Calculations on OR-charges in the U.S. revealed costs ranging from USD 21.80 to 133.12 per OR-minute depending on complexity of performed surgeries [23,24]. In our study, a single OR-minute has been calculated being available at a price of EUR 65 yielding that ASP in NSF procedures is not economically profitable with a confidence of 95% (Figure 4). The 95% CI was chosen for calculations, because reimbursements in countries using DRG based remuneration regulatory systems are not predictable in their total extent. They rather depend on an array of factors including specific individual secondary diagnoses, length of hospital stay, conditions causing interruption of post-surgical fracture-related inpatient treatment in favor of concurrent therapies that may necessitate patient transfer to other clinics/departments, etc. All these factors can extend or cut down the standardized reimbursement volume regulated by the DRG system. Moreover, changes in reimbursements or diagnosis groupings result in annually varying remuneration for the surgical treatment discussed here. Presented results are therefore subject to unpredictable medico-economic changes, which is a limitation to long-term validity of study results.

From an economic standpoint, this investigation currently seems to favor ASP in NSF procedures, if prices for implants and/or extended surgical time-demand ranged below certain thresholds. This might theoretically be attainable by improving the economic environment (purchasing implants and materials at lower prices, refining perioperative and surgical workflow through enhanced effectiveness, etc.). However, despite associated higher remuneration, increased accompanying resource consumption did not justify ASP that was—under economical aspects—classified as redundant in the comparison of investigated therapeutic options. In the context of medically equivalent differential indications, such findings are crucial for the decision-making process that is increasingly subjected to economic considerations. To generalize these findings and improve applicability for institutions with differing resource costs or for those in countries not using the DRG system, the following equation calculated from presented results might help to decide whether ASP is economically reasonable in the respective setting:

RD = 0.003 (MC_ASP ×_ ST_ASP_ × TC_OR_)/ΔRB_ASP_, with RD indicating resource dissipation; MC_ASP_, material costs for ASP; ST_ASP_, surgical time for ASP (in minutes); TC_OR_, time costs per OR minute; ΔRB_ASP_, reimbursement difference between procedures with or without ASP.

Thus, RD values >1 indicate economic disadvantages, RD values < 1 indicate economic advantages (Figure 5).

Furthermore, although complications associated with ASP were reported being asymptomatic [1], severe events are described in the literature [25,26,27]. In order to reduce the risk of neural compression caused by cement-leakage through transforaminal or central fracture gaps, therapeutic options were restricted to NSF without ASP, if transforaminal or central fractures were diagnosed in pre-operative CT scans. This non-random allocation to study groups (concerning 17.3% of participants) represents another limitation to the study. However, undifferentiated randomization of participants to treatment groups would have implicated performing ASP despite pre-surgical identification of transforaminal and/or central fracture patterns. This might have provoked events of cement leakage into neural foramina or into the spinal canal with potentially hazardous consequences for participants’ health concerning integrity and function of neural structures responsible for urinary/defecation continence, and sensorimotor function of the lower extremities. Therefore, only subjects with exclusively transalar fracture patterns (identified in 82.7% of participants) had been randomized to both treatment options in underlying study collectives resulting in significantly different rates of sacral fracture patterns (*p* < 0.001), but similar remaining baseline conditions. In this prospective investigation, study result interpretation led to rejection of the hypothesis “NSF is inferior to the combination of NSF + ASP regarding economic aspects”, and confirmed non-inferiority of NSF in terms of procedural profitability.

## 5. Conclusions

In a prospective monocentric clinical study including 52 patients who underwent 3D-image-guided trans-sacral and/or iliosacral screw fixation with or without additional sacroplasty for immobilizing, non-displaced or minimally displaced sacral fragility fractures, significant differences were found between study groups for operative time (*p* ≈ 0.011), fluoroscopy time (*p* ≈ 0.004), implant costs (*p* < 0.001), and reimbursement (*p* ≈ 0.006). For procedural 2D-radiation dose (*p* ≈ 0.236), or 3D-radiation dose (*p* ≈ 0.546), differences between groups were statistically insignificant.

Although comparison of costs and reimbursements indicated a positive financial balance in procedures with additional sacroplasty, profitability was not confirmed, as financial expense for extended operative time prevented an economic advantage in these procedures.

To allow similar economical decisions in health care systems different from the one in this study, or in institutions with differing resource costs, a mathematical formula was developed based on presented study data.

## Figures and Tables

**Figure 1 jcm-11-06136-f001:**
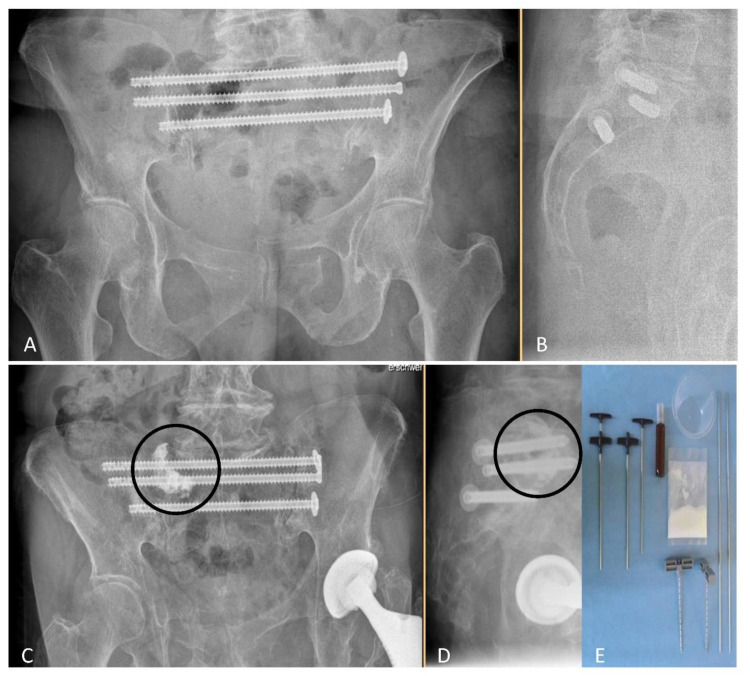
Post-surgical radiographs (anteroposterior (**A**,**C**) and lateral views (**B**,**D**) in standing position) after NSF (**A**,**B**), or NSF + ASP ((**C**,**D**), cement location indicated by circles). Required disposable materials for an ASP procedure are shown in (**E**). NSF indicates navigation-assisted screw fixation; ASP, additional sacroplasty.

**Figure 2 jcm-11-06136-f002:**
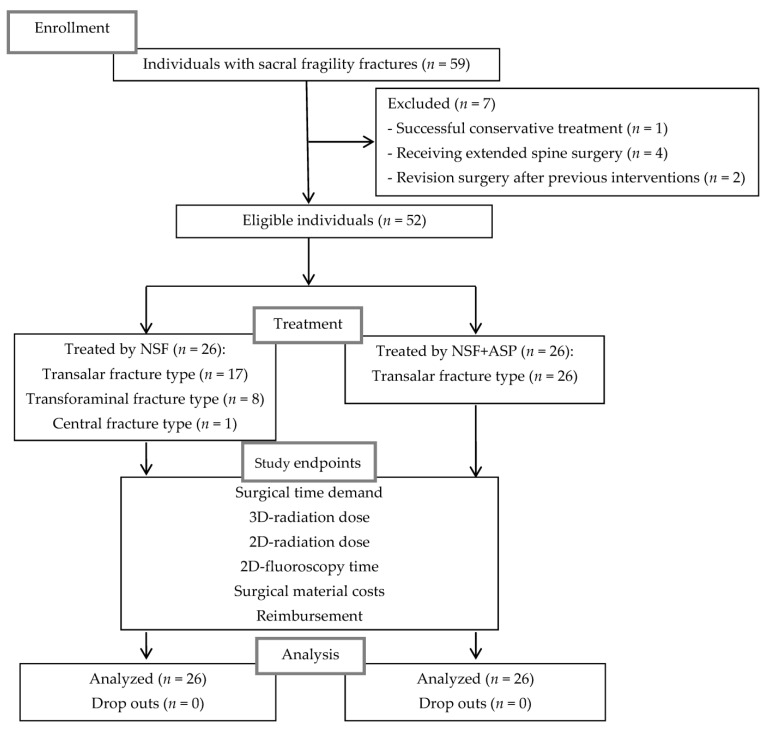
Flow diagram of the study. NSF indicates navigation-assisted screw fixation; ASP, additional sacroplasty.

**Figure 3 jcm-11-06136-f003:**
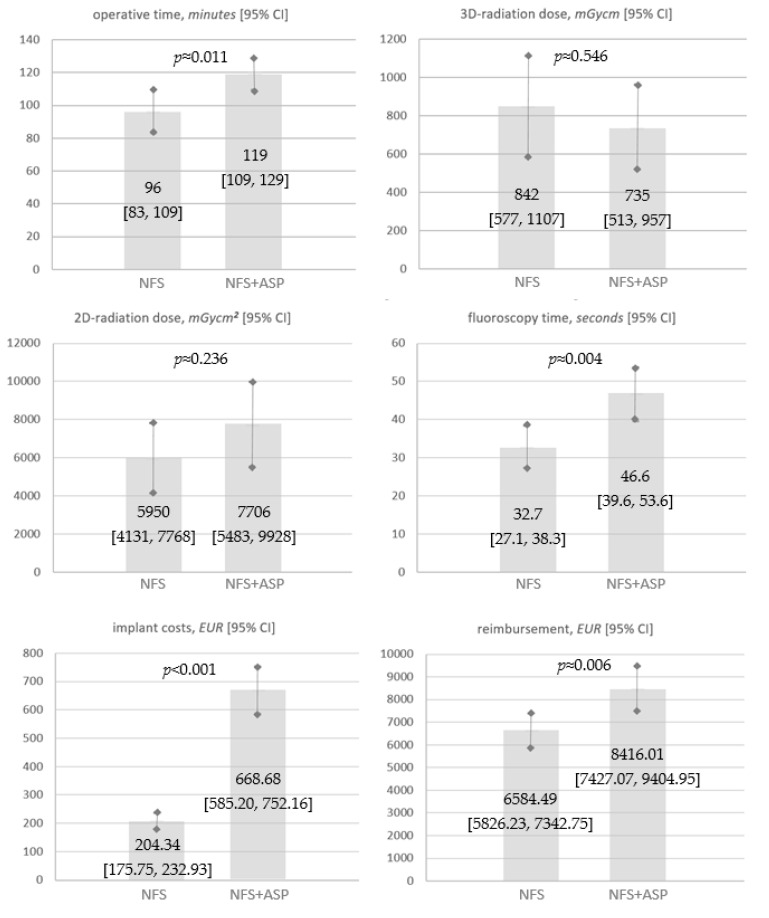
Study endpoints showing no statistically significant differences for 3D- and 2D-radiation dose between groups. Statistically significant differences were found for operative time (*p* ≈ 0.011), fluoroscopy time (*p* ≈ 0.004), procedural costs (*p* < 0.001), and reimbursement (*p* ≈ 0.006). Statistical significance was assumed at a *p* < 0.05 (Student’s *t*-test). NSF indicates navigation-assisted screw fixation; ASP, additional sacroplasty; 2D, two-dimensional; 3D, three-dimensional; 95% CI, 95% confidence interval; mGycm, milligray centimeter.

**Figure 4 jcm-11-06136-f004:**
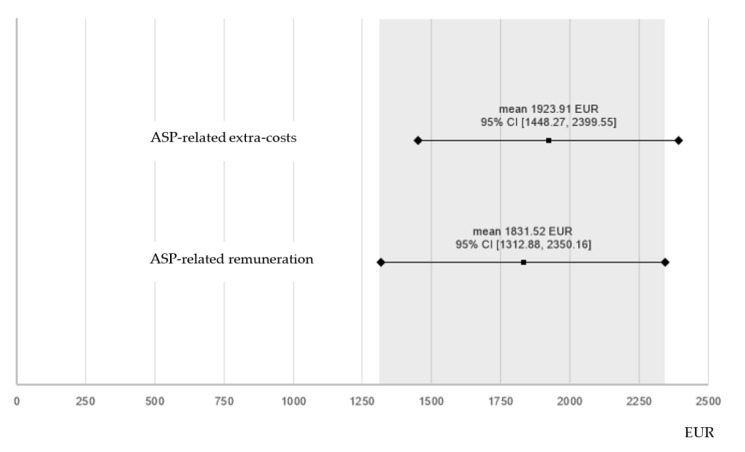
Average remuneration for ASP (EUR 1831.52 [difference between average reimbursements for NSF + ASP and NSF]) is slightly below mean sacroplasty-related extra-costs for implants/devices and extended surgical time (EUR 1923.91). The 95% CI of expenditure for ASP overlaps the upper margin of the 95% CI of sacroplasty-related remuneration (grey field), indicating almost cost-coverage, but non-profitability of ASP. NSF indicates navigation-assisted screw fixation; ASP, additional sacroplasty; 95% CI, 95% confidence interval.

**Figure 5 jcm-11-06136-f005:**
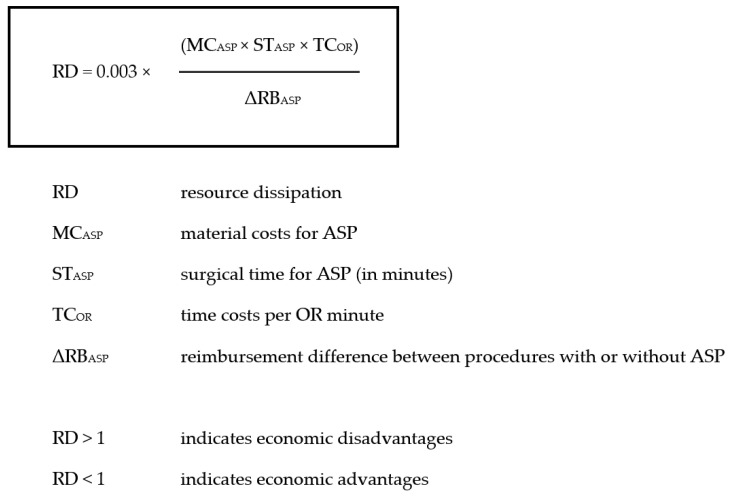
Mathematical formula for the decision-making process to calculate whether ASP is economically reasonable in a certain economic environment. ASP indicates additional sacroplasty.

**Table 1 jcm-11-06136-t001:** Inclusion and exclusion criteria of the study.

Inclusion criteria	- non-/minimally displaced sacral fracture AND - failed conservative therapy of at least 14 days comprising mobilization and pain medication including opioids AND - visual analogue scale (VAS) pain level > 5 AND - Oswestry Disability Index (ODI) score > 40
Exclusion criteria	- successful mobilization with crutches or walkers during conservative treatment OR - identification of pain source different from the sacrum, e.g., the lumbar spine, OR - having undergone previous sacral intervention in form of sacroplasty or sacral screw fixation

For including an individual to the study, all inclusion criteria had to be fulfilled. For exclusion of a subject, presence of a single exclusion criterion sufficed.

**Table 2 jcm-11-06136-t002:** Baseline characteristics. Significance was reached at a *p* < 0.05 (Student’s *t*-test for continuous variables, *chi^2^*-test for categorical variables), significant values are in bold.

Characteristic	NSF (*n* = 26)	NSF + ASP (*n* = 26)	*p*
Age, years—mean (range)	75 (36–90)	77 (55–89)	0.392
Sex women men	23 3	25 1	0.610
Numbers of sacral fracture sites ^a^ in S1 and S2	60	64	1.000
S1/S2 fracture sites ^a^	44/16	43/21	0.556
- transalar fracture type	42	64	**<0.001**
- transforaminal fracture type	17	0	**<0.001**
- central fracture type	1	0	0.484
Numbers of screws inserted in S1/in S2	50/8	55/7	0.679

NSF indicates navigation-assisted screw fixation; ASP, additional sacroplasty; S1, sacral vertebra I; S2, sacral vertebra II; ^a^ fracture sites were separated in “S1, right side”, “S1, center”, “S1, left side”, “S2, right side”, “S2, center”, “S2, left side”.

**Table 3 jcm-11-06136-t003:** Procedural costs regarding screws inserted (depending on screw dimensions), and sacroplasties performed.

Costs of…	Number of Items		Expenses (EUR)	
	NSF	NSF + ASP	NSF	NSF + ASP
…screws [diameter (mm) × length (mm)] ^a^	58	62		
7.3 × 75–95, long thread (DePuy Synthes)	8	4	473.60	236.80
7.3 × 100–130, long thread (DePuy Synthes)	6	14	385.20	898.80
7.3 × 135–150, long thread (DePuy Synthes)	15	16	994.50	1060.80
7.3 × 75–95, short thread (DePuy Synthes)	1	2	58.10	116.20
7.3 × 100–120, short thread (DePuy Synthes)	1	-	63.00	-
7.3 × 125–130, short thread (DePuy Synthes)	1	-	64.80	-
7.3 × 135–150, short thread (DePuy Synthes)	18	12	1180.80	787.20
7.5 × 75–115, full thread (Königsee)	2	-	218.60	-
7.5 × 120–165, full thread (Königsee)	-	12	-	1611.60
7.5 × 75–115, short thread (Königsee)	5	2	533.75	213.50
7.5 × 120–165, short thread (Königsee)	1	-	131.75	-
…guidewires for screw insertion (DePuy Synthes)	50	48	875.00	840.00
…guidewires for sacroplasty (DePuy Synthes)	-	32	-	560.00
…guidewires for screw insertion (Königsee)	8	14	182.40	319.20
…guidewires for sacroplasty (Königsee)	-	8	-	182.40
…washers (DePuy Synthes)	45	40	130.50	116.00
…washers (Königsee)	8	13	20.88	33.93
…sacroplasty cement application tools	-	26	-	6763.13
…sacroplasty cement packages	-	26	-	3646.16
Cumulative costs, EUR			5312.88	17,385.72
Mean costs per surgery, EUR			204.34	668.68
Mean osteosynthesis costs per surgery, EUR			204.34	239.77
Mean sacroplasty costs per surgery, EUR			-	428.91

NSF indicates navigation-assisted screw fixation; ASP, additional sacroplasty; mm, millimeter. ^a^ DePuy Synthes Cannulated Screw System; screws have a diameter of 7.3 mm (DePuy Synthes, Zuchwil, Switzerland), Königsee screws have a diameter of 7.5 mm (TIS^TM^ Königsee Implantate GmbH, Allendorf, Germany).

**Table 4 jcm-11-06136-t004:** Summary of results. Differences between groups were significant as to surgical time, fluoroscopy time, implant costs, and reimbursement. Significance was reached at *p* < 0.05 (*Student’s t*-test), significant values are in bold.

Result	NSF (*n* = 26)		NSF + ASP (*n* = 26)		*p*
	Mean (Range)	95% CI	Mean (range)	95% CI	
Surgical time (minutes)	96 (47–167)	[83, 109]	119 (81–211)	[109,129]	**0.011**
3D-radiation dose (mGycm)	842 (120–2745)	[577, 1107]	735 (95–2059)	[513,957]	0.546
2D-radiation dose (mGycm^2^)	5950 (818–25,332)	[4131, 7768]	7706 (2497–24799)	[5483,9928]	0.236
Fluoroscopy time (seconds)	32.7 (7.4–60.3)	[27.1, 38.3]	46.6 (17.7–102.3)	[39.6, 53.6]	**0.004**
Implant costs (EUR/surgery)	204.34 (79.60–421.48)	[175.75, 232.93]	668.68 (514.69–1333.93)	[585.20, 752.16]	**<0.001**
Osteosynthesis costs (EUR/surgery)	204.34 (79.60–421.48)	[175.75, 232.93]	239.77 (163.40–479.13)	[198.31, 281.23]	0.174
Sacroplasty costs (EUR/surgery)	-	-	428.91 (351.29–854.80)	[377.72, 480.10]	**<0.001**
Reimbursement (EUR/surgery)	6584.49 (3507.54–13,144.47)	[5826.23, 7342.75]	8416.01 (4609.61–13456.99)	[7427.07, 9404.95]	**0.006**

NSF indicates navigation-assisted screw fixation; ASP, additional sacroplasty; 95% CI, 95% confidence interval; 2D, two-dimensional; 3D, three-dimensional; mGycm, milligray centimeter.

## Data Availability

The data presented in this study are available within the article and its supplementary materials. A comprehensive data overview is given in Appendix A.

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
