# Peer review of "Resource Consumption and Remuneration Aspects in Navigated Screw Fixation Procedures with or without Additional Sacroplasty for Fragility Fractures of the Sacrum—A Prospective Clinical Study"

_jcm, 2022, doi:10.3390/jcm11206136_

Round 1
Reviewer 1 Report
The authors present a clear description of the problem and performed a clear analysis. The article is well written. There are only minor changes needed, especially concerning tables and figures.
Please use p=X.XXX for indicating p-values
Please use same number of decimal places for all p-values (especially table 4)
The introduction is very well written and expresses the problem clearly. The hypothesis is stated and clear.
M&M:
Don’t use “bolt” markings (lines 76-82)
Please state the name of the hospital where the study was performed (mulitcenter? Monocenter?)
Please revise figure 1. A,B, D is not shown on the pictures.
Figure 2: The revision surgery is not stated explicitely in excluded patients. Costs are missing.
Table 3: Please change € to EUR
Table 4: Cancel the dot behind 0.24
Figure 3: Please include the 95% CI to the graphs
Figure 4: Please check image quality and revise
Please create a figure showing your created formula in lines 303. Please indicate clearly all contents of the formula in the figure. Please also state in the abstract that you created a formula for economical decision in different health systems.
Author Response
Response to Reviewer 1 Comments:
Point 1: The authors present a clear description of the problem and performed a clear analysis. The article is well written. There are only minor changes needed, especially concerning tables and figures.
Please use p=X.XXX for indicating p-values
Please use same number of decimal places for all p-values (especially table 4)
Response 1: All p-values [in the abstract (lines 25-29), in the main text (lines 255-275), in Table 2 and Table 4] are now displayed with the same number of decimal places (in the form p=X.XXX).
Point 2: The introduction is very well written and expresses the problem clearly. The hypothesis is stated and clear.
M&M:
Don’t use “bolt” markings (lines 76-82)
Response 2: There were no bold markings intended in (former) lines 76-82. It seems that these markings have been accidentally inserted while formatting the original file during the submission process. Unfortunately, we are not able to prevent this from happening as changes occurred after or during the file upload.
Point 3: Please state the name of the hospital where the study was performed (multicenter? Monocenter?)
Response 3: This monocentric study was performed in the “Orthopaedische Fachklinik Schwarzach”, Dekan-Graf-Str. 2-6, 94374 Schwarzach, Germany. We have included this information in the “Materials and Methods” section, lines 111-113. After having finished this study, we had registered the study protocol a posteriori in the German Clinical Trials Register (DRKS.de) with the intention to gather more data during our future performances of the index procedure. Therefore, the study protocol is listed on www.drks.de as multicenter study with additional participating clinics.
Point 4: Please revise figure 1. A, B, D is not shown on the pictures.
Response 4: The image annotations had disappeared while formatting the original file during the submission process. We modified the format of the image and saved it as a pdf-file in order to avoid data changes during file upload.
Point 5: Figure 2: The revision surgery is not stated explicitly in excluded patients. Costs are missing.
Response 5: Figure 2 has been reformatted in the text. The original content of the boxes has been truncated due to incorrect formatting, which has partly masked the actual content of the original text in the figure’s boxes. Thanks to the reviewer, this shortcoming has been detected and corrected.
Point 6: Table 3: Please change € to EUR
Table 4: Cancel the dot behind 0.24
Response 6: Table 3 has been corrected according to the reviewer’s proposal. Beside that, the € symbol has also been exchanged by “EUR” in the main text, in Tables 3 and 4, and in Fig. 4.
In Table 4, the dot behind the p-value has been erased.
Point 7: Figure 3: Please include the 95% CI to the graphs
Figure 4: Please check image quality and revise
Response 7: In Figure 3, the 95% CIs are now included in the graphs, and units have been added to the graphs.
Figure 4 has been revised to enhance the image quality. The €-symbol has also been changed into “EUR”.
Point 8: Please create a figure showing your created formula in lines 303. Please indicate clearly all contents of the formula in the figure. Please also state in the abstract that you created a formula for economical decision in different health systems.
Response 8: A figure showing the formula from the text including all contents of the formula has been created. In the abstract, a statement has been inserted that mentions this formula’s potential meaning for economical decisions in other health care systems.

Reviewer 2 Report
1. The abstract does not give a clear definition of the paper's goal.
2. There should be more graphical representation or figures with technical explanation in the Results and Discussion portion of the abstract, which should also include the novelty of this work.
3. The introduction part is very brief and ought to include some reliable references. The literature review needs to include a few up-to-date references from 2017 to 2022.
4. Conclusion section is missing. I recommend adding a conclusion section.
5. Figures caption always comes below it. Change in the manuscript.
6. Explain more about the student’s t-test and chi2 test.
Author Response
Response to Reviewer 2 Comments:
Point 1: The abstract does not give a clear definition of the paper's goal.
Response 1: The paper’s goal has been defined more concretely now: “…This investigation highlights procedural economic aspects and evaluates results with regard to resource scarcity in order to be able to decide, whether ASP has a justification in NSF procedures beyond clinical aspects…“
Point 2: There should be more graphical representation or figures with technical explanation in the Results and Discussion portion of the abstract, which should also include the novelty of this work.
Response 2: We added the phrase “…A formula was developed based on presented study data to allow similar economical decisions in other health care systems or institutions with differing resource costs…” to the Results and Discussion section of the abstract to illustrate the novelty of this work. A corresponding figure is given in the main text (Figure 5).
Point 3: The introduction part is very brief and ought to include some reliable references. The literature review needs to include a few up-to-date references from 2017 to 2022.
Response 3:.The introduction has been augmented by some recent references fitting the topic of reimbursement and economic effects of various surgical and conservative treatment options for sacral fractures. The reference list has accordingly been updated.
Point 4: Conclusion section is missing. I recommend adding a conclusion section.
Response 4: We added a Conclusion section to the end of the main text: “Conclusion: In a prospective mono-centric clinical study including 52 patients who underwent 3D-image-guided transsacral and/or iliosacral screw fixation with or without additional sacroplasty for immobilizing, non-displaced sacral fragility fractures, significant differences were found between study groups for operative time (p≈0.011), fluoroscopy time (p≈0.004), implant costs (p<0.001), and reimbursement (p≈0.006). For procedural 2D-radiation dose (p≈0.236), or 3D-radiation dose (p≈0.546), differences between groups were statistically insignificant.
Although comparison of costs and reimbursements indicated a positive financial balance in procedures with additional sacroplasty, profitability was not confirmed, as financial expense for extended operative time prevented an economic advantage in these procedures.
To allow similar economical decisions in health care systems different from the one in this study, or in institutions with differing resource costs, a mathematical formula was developed based on presented study data.“
Point 5: Figures caption always comes below it. Change in the manuscript.
Response 5: The changes were performed. Figure captions are now below the figures.
Point 6: Explain more about the student’s t-test and chi2 test.
Response 6: We extended the statistical analysis section: “…The Student’s t-test was used for comparing continuous variables following a normal distribution, and the chi2-test for categorical variables. Only if sample sizes were small (n<10), or contingency tables showed a very unequal distribution, the Fisher’s exact test was employed…”
